# ENHANCING SELF-SUPERVISED DEPTH ESTIMATION THROUGH CAMERA PARAMETER PRIORS

## ABSTRACT

Depth estimation is a key topic in the field of computer vision. Self-supervised monocular depth estimation offers a powerful method to extract 3D scene information from a single camera image, allowing training on arbitrary image sequences without the need for depth labels. However, monocular unsupervised depth estimation still cannot address the issue of scale and often requires ground-truth depth data for calibration. In the deep learning era, existing methods primarily rely on relationships between images to train unsupervised neural networks, often overlooking the foundational information provided by the camera itself. In fact, based on physical principles, the camera's intrinsic and extrinsic parameters can be used to calculate depth information for the ground and related areas and extend it from planar regions to full scene depth. To make full use of scene depth, even in the presence of errors, we introduce a contrastive learning self-supervised framework. This framework consists of two networks with the same structure: the Anchor network and the Target network. The predictions from the Anchor network are used as pseudo-labels for training the Target network. Depth reliability is determined by entropy, dividing the predicted depth into positive and negative samples to maximize the use of physical depth information, and effectively enhance the depth estimation accuracy.

## 1 INTRODUCTION

Monocular depth estimation plays a critical role in fields such as computer vision Newcombe et al. (2011); Luo et al. (2021); Tateno et al. (2017), scene understanding Hazirbas et al. (2017), and 3D mapping Li et al. (2023). Its goal is to infer depth from a single RGB image, but this is inherently an ill-posed problem due to scale ambiguity, as the same 2D image can be projected from infinitely many 3D scenes. The advent of convolutional neural networks has significantly advanced monocular depth estimation Simonyan & Zisserman (2014); Szegedy et al. (2015); He et al. (2016), with the most accurate results being achieved through supervised learning Eigen et al. (2014); Fu et al. (2018); Ranftl et al. (2020); Bhat et al. (2021), which requires sparse depth data collected by sensors like LiDAR as labels. The high cost of data collection and labeling has driven researchers to explore self-supervised depth estimation frameworks. Early self-supervised methods used regression modules to estimate per-pixel depth and infer 3D structures Godard et al. (2019); Gordon et al. (2019); Peng et al. (2021); Watson et al. (2019), relying on photometric consistency loss for model training. However, the accuracy of self-supervised monocular depth estimation still falls short when compared to supervised learning methods. In deep learning-driven depth estimation, the rich information provided by sensors is often overlooked. This paper proposes a camera model that combines image semantics with the physical model of the camera (including intrinsic and extrinsic parameters) to calculate the depth information of road surfaces, extending this to the depth of objects on the ground, such as buildings and vehicles. By filling in missing points, we generate a dense depth map, thus providing supervision for self-supervised models without relying on additional equipment.

To effectively utilize the physics depth information, we designed a self-supervised network framework based on contrastive learning. If only the accurate ground areas are selected as pseudo ground truth in the physics depth, many pixels may go unused. We believe that every pixel is essential for model training, even when errors are present. Unreliable predictions may confuse adjacent depth intervals and be used as negative samples for the least likely depth categories. We separate reliable and unreliable pixels based on entropy and utilize all reliable pixels to train the model. Considering

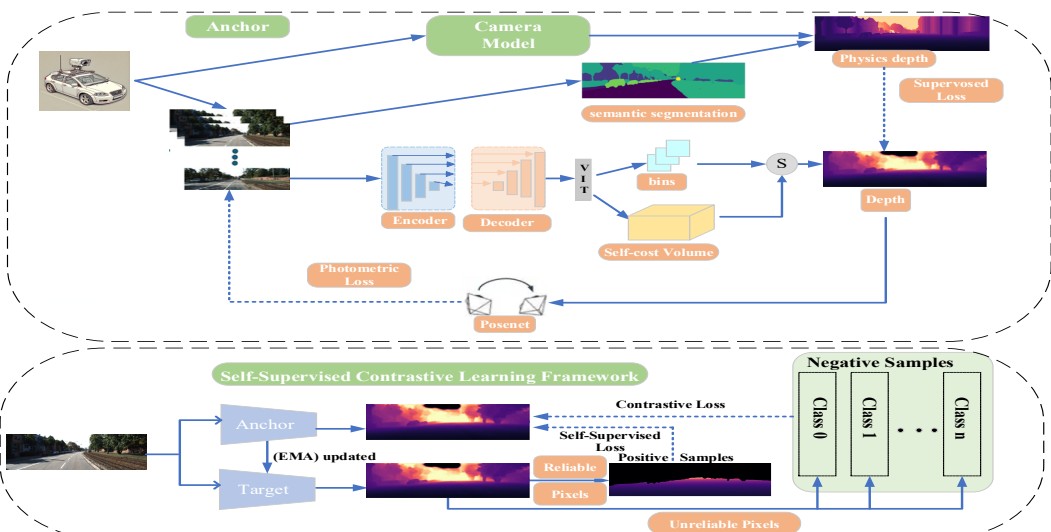

Figure 1: The overview of our framework. The Anchor network uses two decoders to output semantic segmentation and depth, combined with the camera model to compute physics depth as labels. The depth estimation decoder outputs bins and cost volume through ViT, which are aggregated into depth values. Combined with the pose estimated by the pose network, photometric reprojection loss is generated, and $\overline{\Lambda}_t$ is calculated through semantic segmentation. When $\overline{\Lambda}_t \geq \theta_\Lambda$, the scene is considered static, and photometric reprojection error is used; otherwise, the scene is dynamic, and masking is applied. In the contrastive learning framework, The anchor and target networks have the same structure, as shown in the figure above. The predictions from the anchor network serve as pseudo-labels for the target network. Unsupervised loss is calculated for reliable pixels, and contrastive loss is applied to make full use of unreliable pixels.

the training process, as the predictions become more accurate, we adaptively adjust the threshold to distinguish between reliable and unreliable pixels.

In summary, our main contributions include: 1. We propose a novel mechanism that leverages camera physical model parameters to calculate scene depth, providing supervisory signals to the depth estimation network. We refer to this depth information as physics depth. 2. To address the scale uncertainty in self-supervised monocular depth estimation, our method provides an absolute scale, rather than a relative scale alone. 3. For the physics depth calculated from the camera model, we designed a contrastive learning self-supervised neural network training framework that integrates physics depth supervision with self-supervised methods. 4. We developed a method to verify and correct the camera-to-ground calibration results.Figure 1 shows the framework for physics depth computation and self-supervised network training.

## 2 RELATE WORK

### 2.1 DEPTH ESTIMATION

Monocular depth estimation has seen significant advancements since the pioneering work by Eigen et al. (2014),. Since then, the field has evolved with improvements in both network architectures and loss functions Laina et al. (2016); Lee et al. (2018); Liu et al. (2015); Miangoleh et al. (2021). Approaches in supervised monocular depth estimation typically revolve around either pixel-wise regression Eigen et al. (2014); Zhao et al. (2021); Ranftl et al. (2021); Huynh et al. (2020) or pixel-wise classification Fu et al. (2018); Diaz & Marathe (2019). While regression predicts continuous depths, it can pose optimization challenges, whereas classification, though easier to optimize, results in discrete depth predictions. Self-supervised depth estimation has gained prominence due to the difficulty in acquiring accurate ground truth data. The seminal work by Zhou et al. (2017) introduced a framework for jointly training depth and pose networks using image reconstruction loss. Subsequent innovations, such as minimum re-projection loss and auto-masking loss by Godard et al. (2019), further advanced the state of the art. Scale ambiguity in monocular Structure-from-Motion (SfM) models, a common challenge, has been addressed by incorporating real-time data like GPS

or camera velocity in works such as Guizilini et al. (2020) and Chawla et al. (2021). These methods rely on photometric consistency for re-projection Wang et al. (2004). In stereo depth estimation, disparity prediction, which is inversely related to depth, plays a crucial role. Garg et al. (2016) introduced self-supervised training of monodepth models with stereo pairs, which was refined by Godard et al. (2017) using left-right consistency and later extended to continuous disparity prediction by Garg et al. (2020). Stereo models predict absolute depth scales, while monocular models typically predict relative depth, requiring calibration with ground truth. Integrating physics-based depth data improves the accuracy of absolute depth predictions, particularly for datasets like KITTI.

## 2.2 GEOMETRIC PRIORIS

Geometric priors have become increasingly important in monocular depth estimation. Among them, the normal constraint Long et al. (2021); Qi et al. (2018) is widely applied, ensuring that the normal vectors of the predicted depths align with those of the ground truth. The piecewise planarity prior Gallup et al. (2010) provides a practical approximation for real-world scenes. Although monocular depth estimation inherently suffers from ambiguity, and while Transformers have improved prediction accuracy, they do not fundamentally address the core error issues in monocular depth estimation. Geometric priors help alleviate some uncertainty, but their overall contribution to solving the problem remains limited. We utilize camera model parameters to compute scene depth directly. The surface normal method Xue et al. (2020); Wagstaff & Kelly (2021) calculates surface normals and estimates camera height through camera parameters, thereby determining the scale factor. However, while these methods focus on using camera parameters to compute scale, they do not consider how to use the camera model as a prior for depth estimation. Our approach offers more accurate and generalizable depth predictions, further improving model performance.

# 3 PHYSICS-INFORMED DEPTH

## 3.1 PHYSICS-INFORMED DEPTH FOR FULL FIELD OF VIEW

This paper presents a monocular depth estimation algorithm that calculates absolute depth by combining camera intrinsic and extrinsic parameters with semantic segmentation. The method uses physics principles to estimate the depth of flat surfaces within the camera's field of view, generating a physics-based depth map under the assumption that all surfaces are ideal planes. Semantic segmentation is then applied to identify planar regions, and the results are extrapolated to adjacent ground and vertical surfaces, with gaps filled using segmentation information and image inpainting techniques. In planar regions, the accuracy is close to that of LiDAR results. Our method uses a pinhole camera model, known for its minimal distortion and real-world applicability. It can be adapted to different camera types with adjustments based on specific characteristics. For each pixel, a unit vector $(\hat{r})$ is computed, representing the camera ray direction, which translates the pixel's position into its line of sight in the physical world $\hat{r} = \frac{[u,v,f]}{\sqrt{u^2+v^2+f^2}}$. The pixel coordinates $(u, v)$ originate from the optical center $(O_x, O_y)$, or principal point. $f$ is the focal length. The scale of the physical depth is derived based on the dimensions of the image; however, the depth map obtained from self-supervised monocular depth estimation often has different dimensions from the image. To use physical depth as a supervisory signal for the self-supervised network, we need to adjust the scale of the physical depth to match the depth predicted by the network. However, directly scaling the physical depth does not align with the principles of the camera model. Therefore, we modify the scale using a camera-based approach, with detailed formulas provided in the supplementary material 1. For a camera with roll, pitch, and yaw angles, the rotation matrix $(R_c)$ representing the camera's orientation relative to the ground as:

$$R_{roll} = \begin{bmatrix} 1 & 0 & 0 \\ 0 & c(roll) & s(roll) \\ 0 & -s(roll) & c(roll) \end{bmatrix}, R_{pitch} = \begin{bmatrix} c(pitch) & 0 & -s(pitch) \\ 0 & 1 & 0 \\ s(pitch) & 0 & c(pitch) \end{bmatrix} \quad (1)$$

$$R_{yaw} = \begin{bmatrix} c(yaw) & s(yaw) & 0 \\ -s(yaw) & c(yaw) & 0 \\ 0 & 0 & 1 \end{bmatrix}, R_c = R_{yaw} * R_{pitch} * R_{roll} \quad (2)$$

Using $R_c$ we rotate the camera ray vector to align it with the ground coordinate system: $\hat{r}_c = R_c * \hat{r}'$ Since $\hat{r}_c(r_{c,u}, r_{c,v}, r_{c,f})$ is a unit vector, the 3D coordinates of the point, $P = (x_c, y_c, z_c)$, on the ground surface in camera's coordinate system can be determined by multiplying $r_c$ with the point-to-point distance $(d)$ of the ground point from camera. $[x_c, y_c] = d * [r_{c,u}, r_{c,v}]$. When the height of the camera $(h)$ is known from the camera's extrinsic parameters and assuming the camera coordinate system's y-axis is oriented downwards, then $y_c = h$, and the point-to-point distance $d$ and $x_c$ can be calculated as shown below: $d = h/r_{c,v}, x_c = d * r_{c,u}$. The projection of a three-dimensional point from the camera coordinate system $(x_c, y_c, z_c)$ to the two-dimensional image plane $(u, v)$, can be accurately represented using the following linear camera model equation:

$$Z_c \begin{bmatrix} u \\ v \\ 1 \end{bmatrix} = \begin{bmatrix} f'_x & 0 & O'_x \\ 0 & f'_y & O'_y \\ 0 & 0 & 1 \end{bmatrix} \begin{bmatrix} x_c \\ y_c \\ z_c \end{bmatrix}, K = \begin{bmatrix} f'_x & 0 & O'_x \\ 0 & f'_y & O'_y \\ 0 & 0 & 1 \end{bmatrix} \tag{3}$$

where **K** denotes the camera's intrinsic matrix. By substituting $x_c$ and $y_c$ into Eq. 6, we can derive $z_c$ for any pixel $(u, v)$ on the ground, allowing depth and 3D coordinate computation for all ground pixels using the camera height. This method was evaluated on the KITTI Geiger et al. (2013) and Cityscapes Cordts et al. (2016).

## 3.2 EXTENSION OF PHYSICS-INFORMED DEPTH

Our physics-based depth method closely aligns with LiDAR data for flat surfaces but may over-fit to road regions. To improve effectiveness in diverse scenes, we extended the method to cover the entire image. By assuming flat surfaces at camera level and incorporating vertical elements like vehicles and buildings, we create a more comprehensive depth map, termed Edge Extended Physics depth. We extend the physics depth to vertical entities in contact with flat surfaces, like vehicles and buildings, by propagating depth values from intersection points, forming the Edge Extended Physics depth. Missing depth for partially connected objects is filled using the Telea inpainting technique Telea (2004). For objects not touching the ground, depth is extrapolated from nearby objects. The sky is filled with 1.5 times the maximum inpainted depth, creating a seamless Dense Physics depth label for subsequent net-

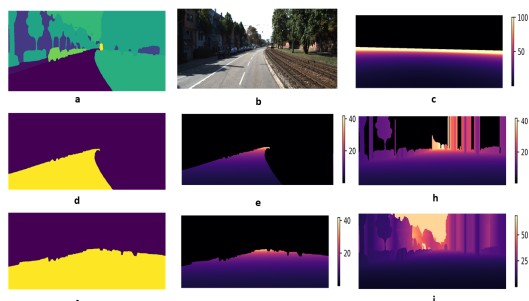

Figure 2: Physics Depth Methodology on KITTI: (a) semantic segmented image (b) RGB image (c) full physics depth (d) road segmented from semantic segmented image (e) physics depth of road (f) ground segmented from semantic segmented image (g) physics depth of ground (h) edge extended physics depth (i) dense physics depth.

works. The effectiveness of Our method has been validated on the KITTI Geiger et al. (2013) and Cityscapes Cordts et al. (2016) datasets, showing accuracy closely aligned with LiDAR-derived depth measurements, particularly for ground surfaces.

Five types of physics depth are analyzed: complete, road, ground, edge-expanded, and dense physics depth. Using the KITTI dataset, Figure 2 illustrates these types. The process starts with applying our segmentation result to segment the image, where 'd' and 'f' refer to road and flat ground areas. The images in 'c', 'e', 'h', 'g', and 'i' show different stages of the physics depth calculation.

## 3.3 CORRECTION OF CAMERA-TO-GROUND CALIBRATION

Camera extrinsic is an important component in calculating the physics depth. Extensive methods have been developed for camera calibration, such as Zhengyou (1998). In our analysis of the Physics Depth Algorithm on the complete KITTI dataset, optimal results were evident on the initial day (2011-09-26). However, performance diminished in the subsequent days. Here, we provide a camera-to-road calibration verification and correction method. Taking KITTI dataset as an example, given the algorithm's excellence on day one, we postulated that discrepancies in the later days might stem from inconsistencies in the KITTI dataset camera calibration parameters, specifically $R_c$.

To substantiate this, we computed the camera calibration rotation matrix $R_c = R_{cl} \times R_{lg}$ using LiDAR data. $R_{cl}$ is the rotation matrix to transition a point from the camera to the LiDAR, and $R_{lg}$ transforms a point from the LiDAR to the ground. Below is the procedure for verifying and correcting the camera-to-road calibration.

**Camera Calibration Correction Methodology:** 1. $R_{lc}$ is provided in KITTI's calibration set. Compute $R_{cl}$ as the transpose of $R_{lc}$. 2. Derive $R_{lg}$ through the following steps: (a.) Project 3D depth points, obtained from LiDAR, onto the LiDAR frame to construct a three-dimensional hyperplane representing the ground surface. (b.) Determine the centroid of these 3D ground points in the LiDAR frame, and decompose the surface normal at this centroid to ascertain roll and pitch angles. (c.) Use the derived angles in the third step of the physics depth to obtain $R_{gl}$, and compute its transpose to yield $R_{lg}$. 3. Combine $R_{cl}$ and $R_{lg}$ from the above steps to get $R_c$. 4. Use $R_c$ in the third step to achieve Camera Calibration Corrected Physics Depth. Here, we are providing a method to verify and correct the camera-to-road calibration output. Through our testing, the camera-to-road calibration in KITTI maintains errors.

## 4 SELF-SUPERVISED CONTRASTIVE DEPTH LEARNING

### 4.1 NETWORK ARCHITECTURE

In our study, selecting the physics depth of ground regions as labels based on accuracy may result in many pixels being unused due to errors. We believe every pixel is crucial for model training, even if its prediction is uncertain. While interpolated depth may cause confusion in similar ranges, it should maintain high confidence for pixels in larger disparity ranges, allowing those pixels to be convincingly treated as negative samples. To fully leverage this data, we developed a self-supervised contrastive learning framework. We discretize depth values and linearly combine the predicted classifications to obtain accurate estimates. We uses physics depth as labels to train an anchor network, and the anchor network's predictions for accurate regions are retained as pseudo-labels for self-supervised training of the target network, progressively increasing the proportion of accurate regions in each iteration.

In this study, the depth regression task is transformed into a classification task by discretizing continuous depth values into fixed-width bins. To improve precision and mitigate depth discontinuities, final depth values are reconstructed through a linear combination of bin centers. Additionally, the Spacing-Increasing Discretization (SID) strategy from Fu et al. (2018) is used to divide the depth range into non-uniform intervals, enhancing accuracy for small depth variations at long distances. $t_i^{\text{SID}} = e^{\log \alpha + i \cdot \frac{\log \beta / \alpha}{n}}, \quad i = 0, 1, \ldots, N$ Here, $t_i \in \{t_0, t_1, \ldots, t_N\}$ represents the discrete depth thresholds. The $N$ Softmax scores $p_k$, where $k = 1, ..., N$, at each pixel are interpreted as probabilities over the depth-bin centers $c(\mathbf{b})$, which are computed from the bin-width vector $\mathbf{b}$ as follows:

$$c(b_i) = d_{min} + (d_{max} - d_{min})(b_i/2 + \sum_{j=1}^{i-1} b_j), \tilde{d} = \sum_{k=1}^{N} c(b_k) p_k \qquad (4)$$

where the final depth value $\tilde{d}$ is calculated from the linear combination of Softmax scores at that pixel and the depth-bin-centers $c(\mathbf{b})$. Our encoder-decoder architecture is based on the transformer structure of MonoVit Zhao et al. (2022), with the semantic segmentation task being trained using supervised learning. For the depth estimation task, we employ self-supervised learning, where the depth decoder receives input from the ViT and predicts depth bins and cost volumes. The cost volume is constructed by comparing relative distances between different points within the image, inspired by SQLDepth Wang et al. (2023b), which introduces coarse query points to calculate object-to-point distances, reducing computational costs. By dividing the feature map into larger patches and enhancing these patch embeddings using a transformer, we implicitly represent objects within the image. The final layer of the ViT outputs a dot product and softmax, which are fed into a multilayer perceptron (MLP) to predict the depth intervals (bins). On each plane of the cost volume, pixel-wise softmax is first applied to convert each plane into a probability map for each pixel. These maps are then used for weighted summation to obtain a vector representing different depth counts. Using the depth intervals extracted from the cost volume, the cost volume is compressed into a volume matching the shape of the depth intervals by applying 1×1 convolutions. The compressed volume is converted into probability maps on the planes, and depth for each pixel is computed through probability-weighted linear combinations, aggregating depth values using the depth interval centers and their corresponding probability weights.

For the supervised training utilizing physics depth as ground truth, we employ the cross-entropy loss function $\mathcal{L}_s$ is cross-entropy (CE) loss:

$$\mathcal{L}_{phy} = \frac{1}{|\mathcal{B}_l|} \sum_{(\mathbf{x}_i^l, \mathbf{d}_i^l) \in \mathcal{B}_l} \ell_{ce}(\mathbf{d}_i^{l'}, \mathbf{d}_i^l), \quad (5)$$

where $\mathbf{d}_i^l$ represents the physics depth for the $i$-th image. For consecutive frames $I_{t-1}$ and $I_t$, our model independently estimates their respective depths, $D_{t-1}$ and $D_t$. These frames are then projected into 3D point clouds, $Q_{t-1}$ and $Q_t$, using the principles in Eq. 6. The camera's motion between these frames is estimated by the pose network, producing a transformation matrix $T_{t-1 \to t}$. This matrix is applied to the point cloud $Q_t$ to generate an estimated point cloud $\hat{Q}_{t-1}$, expressed as $\hat{Q}_{t-1} = T_{t-1 \to t} Q_t$. The image $I_t$ is then reconstructed by warping the previous frame $I_{t-1}$ according to Eq. 7. The photometric loss is subsequently calculated using Eq. 7, by comparing the reconstructed image $\hat{I}_{t-1 \to t}$ with the actual target image $I_t$.

$$Q_{t-1}^{xy} = D_{t-1}^{xy} \cdot K^{-1} \begin{bmatrix} x \\ y \\ 1 \end{bmatrix} \quad (6)$$

$$I_{t-1 \to t}[u] = I_{t-1} \langle u' \rangle, L_{ph} = ph(I_t, I_{t-1 \to t}) \quad (7)$$

$$ph(I_t, I_{t-1 \to t}) = \frac{\alpha}{2}(1 - SSIM(I_t, I_{t-1 \to t})) + (1 - \alpha) \|(I_t, I_{t-1 \to t})\|_1 \quad (8)$$

Here, $\alpha$ is set to 0.85 Godard et al. (2019), and $ph$ represents the photometric reconstruction error.

$$L_{ph}(p) = \min_{s \in [-1,1]} pe(I_{t-1}(p), I_{t-1 \to t}(p)) \quad (9)$$

1 stands for forward, 2 stands for backward.

$$L_s = |\partial_x d_t^*| e^{-|\partial_x I_t|} + |\partial_y d_t^*| e^{-|\partial_y I_t|} \quad (10)$$

Following Godard et al. (2019), we use edge smoothness loss to sharpen edges and depth surfaces.

## 4.2 SELF-SUPERVISED CONTRASTIVE LEARNING

We use a contrastive learning self-supervised framework, as shown in Fig 1, where the Anchor Network and Target Network share the same architecture. The only difference between the two models is how their weights are updated. The architecture and weights $\theta_s$ of the Anchor Network follow Section 4.1, while the weights $\theta_s$ of the Target Network are updated as the exponential moving average of the Anchor Network's weights. We use physics depth as labels to train the Anchor model, while simultaneously updating the Target model. For the depth predicted by the Target model, we ignore unreliable pseudo-label pixel locations when calculating the unsupervised loss and use contrastive loss to fully leverage the unreliable pixels excluded from the unsupervised loss. To mitigate overfitting to low-quality pseudo-labels of physics depth, we filter out unreliable labels based on the entropy of each pixel's probability distribution. Specifically, let $\mathbf{p}_{ij}$ represent the softmax probabilities produced by the Target model for the $i$-th unlabeled image at pixel $j$, where $C$ denotes the number of classes. Its entropy is computed by:

$$\mathcal{H}(\mathbf{p}_{ij}) = -\sum_{c=0}^{C-1} p_{ij}(c) \log p_{ij}(c), \quad (11)$$

where $p_{ij}(c)$ represents the value of $\mathbf{p}_{ij}$ at the $c$-th dimension. We classify pixels with entropy in the top $\alpha_t$ at training epoch $t$ as unreliable pseudo-labels. These unreliable labels are excluded from supervision. We define the pseudo-label for the $i$-th unlabeled image at pixel $j$ as:

$$\hat{d}_{ij}^u = \begin{cases} \arg\max_c p_{ij}(c), & \text{if } \mathcal{H}(\mathbf{p}_{ij}) < \gamma_t, \\ \text{ignore}, & \text{otherwise}, \end{cases} \quad (12)$$

where $\gamma_t$ represents the entropy threshold at $t$-th training step. The setting of $\gamma_t$ is based on Wang et al. (2022). As self-supervised training progresses, the predicted depth in unlabeled regions becomes more reliable, allowing a gradual reduction in the proportion of unreliable pixels. Once reliable pseudo-labels are obtained, they are included in the unsupervised loss in Eq. 13. For self-supervised training with pseudo-labeled images, we use cross-entropy loss $\mathcal{L}_u$.

$$\mathcal{L}_u = \frac{1}{|\mathcal{B}_u|} \sum_{\mathbf{x}_i^u \in \mathcal{B}_u} \ell_{ce}(\hat{\mathbf{d}}_u^{l'}, \hat{\mathbf{d}}_i^u), \quad (13)$$

| Method | Scale | Test | AbsRel ↓ | Sq Rel↓ | RMSE↓ | RMSElog ↓ | $\delta < 1.25$ ↑ | $\delta < 1.25^2$ ↑ | $\delta < 1.25^3$ ↑ |
|---|---|---|---|---|---|---|---|---|---|
| Monodepth2 Godard et al. (2019) | LiDAR Scale | 32.260 | 0.159 | 1.689 | 5.168 | 0.238 | 0.830 | 0.931 | 0.967 |
| | Physics Depth Scale | 32.487 | 0.158 | 1.968 | 5.287 | 0.242 | 0.842 | 0.930 | 0.966 |
| MonoVit Zhao et al. (2022) | LiDAR Scale | 28.354 | 0.110 | 0.759 | 4.248 | 0.199 | 0.872 | 0.954 | 0.979 |
| | Physics Depth Scale | 28.096 | 0.108 | 0.743 | 4.241 | 0.200 | 0.874 | 0.955 | 0.979 |
| SQLDepth Wang et al. (2023b) | LiDAR Scale | 43.51 | 0.087 | 0.659 | 4.096 | 0.165 | 0.920 | 0.970 | 0.984 |
| | Physics Depth Scale | 44.17 | 0.089 | 0.664 | 4.101 | 0.169 | 0.918 | 0.969 | 0.982 |

Table 1: Evaluation of models with LiDAR Depth Scaling Factor and Physics Depth Scaling Factor.

where $\hat{\mathbf{d}}_i^u$ is the pseudo-label for the $i$-th unlabeled image. The weight $\lambda_u$ for $\mathcal{L}_u$ is defined as the reciprocal of the percentage of pixels with entropy smaller than threshold $\gamma_t$ in the current mini-batch multiplied by a base weight $\eta$:

$$\lambda_u = \eta \cdot \frac{|\mathcal{B}_u| \times H \times W}{\sum_{i=1}^{|\mathcal{B}_u|} \sum_{j=1}^{H \times W} \left[ \hat{y}_{ij}^u \neq \text{ignore} \right]} \quad (14)$$

where $\neq$ is the indicator function, and $\eta$ is set to 1. Since physics depth is accurate in flat regions, errors elsewhere may lead to inaccurate pseudo-labels from the Anchor network. Ignoring these areas would reduce the amount of available training data. However, unreliable physics depth is classified as less likely to belong to regions with large depth differences, so we select it as a negative sample. Our contrastive learning framework consists of three components: anchor pixel, positive candidate, and negative candidate. During training, anchor pixels are sampled for each class in the mini batch. The set of features for labeled anchor pixels in class $c$ is denoted as $\mathcal{A}_c^l$:

$$\mathcal{A}_c^l = \{ \mathbf{z}_{ij} \mid d_{ij} = c, p_{ij}(c) > \delta_p \}, \quad (15)$$

where $d_{ij}$ is the ground truth for pixel $j$ in labeled image $i$, $\mathbf{z}_{ij}$ represents its feature, and $\delta_p$ is the positive threshold, set to 0.3. For unlabeled data, $\mathcal{A}_c^u$ is similarly defined using the pseudo-label $\hat{d}_{ij}$, and the final set of all qualified anchors for class $c$ is denoted as $\mathcal{A}_c$.

$$\mathcal{A}_c^u = \left\{ \mathbf{z}_{ij} \mid \hat{d}_{ij} = c, p_{ij}(c) > \delta_p \right\}, \mathcal{A}_c = \mathcal{A}_c^l \cup \mathcal{A}_c^u. \quad (16)$$

**Positive and Negative Samples.** For each class, the positive sample is represented by the centroid of all anchors, computed as:

$$\mathbf{z}_c^+ = \frac{1}{|\mathcal{A}_c|} \sum_{\mathbf{z}_c \in \mathcal{A}_c} \mathbf{z}_c. \quad (17)$$

The negative samples are determined using a binary variable $n_{ij}(c)$, which indicates if the $j$-th pixel of image $i$ qualifies as a negative sample for class $c$. This is defined as:

$$n_{ij}(c) = \begin{cases} n_{ij}^l(c), & \text{if image } i \text{ is labeled}, \\ n_{ij}^u(c), & \text{otherwise}, \end{cases} \quad (18)$$

For labeled images, a pixel qualifies as a negative sample for class $c$ if: (a) it does not belong to class $c$, and (b) it is difficult to distinguish between class $c$ and its true category. This is represented by:

$$n_{ij}^l(c) = [y_{ij} \neq c] \cdot [0 \leq \mathcal{O}_{ij}(c) < r_l], \quad (19)$$

where $\mathcal{O}_{ij}$ represents the pixel-level category ranking, and $r_l$ is the lower rank threshold, set to 3. For unlabeled images, a pixel is considered a negative sample for class $c$ if: (a) it is unreliable, (b) it is unlikely to belong to class $c$, and (c) it does not belong to the least probable categories.

$$n_{ij}^u(c) = [\mathcal{H}(\mathbf{p}_{ij}) > \gamma_t] \cdot [r_l \leq \mathcal{O}_{ij}(c) < r_h], \quad (20)$$

where $r_h$ is the upper rank threshold set to 20. Finally, the set of negative samples for class $c$ is:

$$\mathcal{N}_c = \{ \mathbf{z}_{ij} \mid n_{ij}(c) = 1 \}. \quad (21)$$

$\mathcal{L}_c$ represents the pixel-level InfoNCE Oord et al. (2018) loss, defined as:

$$\mathcal{L}_c = - \frac{1}{C \times M} \sum_{c=0}^{C-1} \sum_{i=1}^{M} \log \left[ \frac{e^{\langle \mathbf{z}_{ci}, \mathbf{z}_{ci}^+ \rangle / \tau}}{e^{\langle \mathbf{z}_{ci}, \mathbf{z}_{ci}^+ \rangle / \tau} + \sum_{j=1}^{N} e^{\langle \mathbf{z}_{ci}, \mathbf{z}_{cij}^- \rangle / \tau}} \right], \quad (22)$$

where $M$ is the total number of anchor pixels, and $\mathbf{z}{ci}$ denotes the representation of the $i$-th anchor for class $c$. Each anchor pixel is associated with one positive sample, $\mathbf{z}{ci}^+$, and $N$ negative samples, $\mathbf{z}_{cij}^-$. The feature representation $\mathbf{z} = g \circ h(\mathbf{x})$ is obtained from the representation head. The cosine similarity between two pixel features, denoted as $\langle \cdot, \cdot \rangle$, ranges from $-1$ to 1, requiring a temperature parameter $\tau$. Following Wang et al. (2022), we set $M = 50$, $N = 256$, and $\tau = 0.5$.

## 5 EXPERIMENT

### 5.1 PHYSICS-INFORMED DEPTH EVALUATION

| | Full Physics depth | Road Surface Physics depth | Flat Surface Physics depth | Edge Extended Physics depth | Dense Physics depth |
|---|---|---|---|---|---|
| +/- 5% error | 47.29% | 80.24% | 60.30% | 41.83% | 38.88% |
| +/- 10 % error | 58.34% | 99.33% | 74.89% | 55.44% | 52.45% |

Table 2: Physics depth in a sample KITTI image.

| Method | Type | Year | Resolution | AbsRel ↓ | Sq Rel↓ | RMSE↓ | RMSE log↓ | $\delta < 1.25$ ↑ | $\delta < 1.25^2$ ↑ | $\delta < 1.25^3$ ↑ |
|---|---|---|---|---|---|---|---|---|---|---|
| Groco Cecille et al. (2024) | M | 2024 | 1024×320 | 0.113 | 0.851 | 4.756 | 0.197 | 0.870 | 0.958 | 0.980 |
| Velocity depth Zhou et al. (2020) | M | 2020 | 1024×320 | 0.112 | 0.816 | 4.715 | 0.190 | 0.880 | 0.960 | 0.982 |
| SelfOcc Huang et al. (2024) | MS | 2024 | 1024×320 | 0.099 | 0.711 | 4.586 | 0.186 | 0.880 | 0.960 | 0.982 |
| HR-Depth Lyu et al. (2021) | MS | 2021 | 1024×320 | 0.101 | 0.716 | 4.395 | 0.179 | 0.899 | 0.966 | 0.983 |
| Lite-Mono Zhang et al. (2023) | M | 2023 | 1024×320 | 0.097 | 0.710 | 4.309 | 0.174 | 0.905 | 0.967 | 0.984 |
| MonoViT Zhao et al. (2022) | M | 2023 | 1024×320 | 0.096 | 0.714 | 4.292 | 0.172 | 0.908 | 0.968 | 0.984 |
| DualRefine Bangunharcana et al. (2023) | MS | 2023 | 1024×320 | 0.096 | 0.694 | 4.264 | 0.173 | 0.908 | 0.968 | 0.984 |
| ManyDepth Watson et al. (2021) | M | 2021 | 1024×320 | 0.087 | 0.685 | 4.142 | 0.167 | 0.920 | 0.968 | 0.983 |
| RA-Depth He et al. (2022) | M | 2022 | 1024×320 | 0.097 | 0.608 | 4.131 | 0.174 | 0.901 | 0.968 | 0.985 |
| PlaneDepth Wang et al. (2023a) | MS | 2023 | 1280×384 | 0.090 | 0.584 | 4.130 | 0.182 | 0.896 | 0.962 | 0.981 |
| SQLDepth Wang et al. (2023b) | M | 2023 | 1024×320 | 0.087 | 0.659 | 4.096 | 0.165 | 0.920 | 0.970 | 0.984 |
| ProDepth Woo et al. (2024) | M | 2024 | 1024×320 | 0.087 | 0.632 | 3.885 | 0.161 | 0.921 | 0.970 | 0.985 |
| Ours | M | 2024 | 1024×320 | 0.085 | 0.583 | 3.770 | 0.158 | 0.922 | 0.970 | 0.986 |

Table 3: The quantitative depth comparison using the Eigen split of the KITTI dataset Geiger et al. (2013). M: trained with monocular videos; MS: trained with stereo pairs.

**Error distribution:** The comparison in Fig. 3 and Table 2 shows that the physics-based depth estimation is highly accurate for road surfaces (b), with over 99% of pixels having less than 10% error and more than 81% having less than 5% error compared to LiDAR data. This suggests that physics-based depth can reliably substitute LiDAR for scaling in self-supervised monocular depth estimation on flat surfaces. However, accuracy decreases when applied to surfaces like sidewalks and parking lots, which are not perfectly level with the camera, and errors increase further when extending the logic to vertical surfaces.

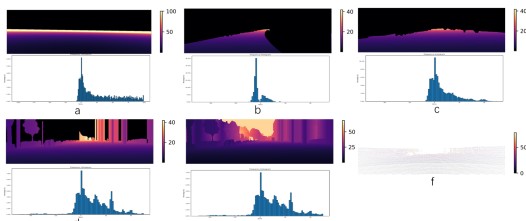

Figure 3: Error distribution of Physics depth: (a) full physics depth and error distribution (b) road physics depth and error distribution (c) flat surface physics depth and error distribution (d) edge extended physics depth and error distribution (e) dense physics depth and error distribution (f) sparse LiDAR depth as ground truth.

**KITTI Dataset Correction:**

The KITTI dataset consists of five distinct calibration files, each corresponding to data collected on different days. In Fig. 6, we conducted a percentage error frequency distribution analysis for each day, and the results are as follows: the error frequency histograms clearly demonstrate a substantial improvement in the performance of the physics depth algorithm after the KITTI camera calibration was corrected using LiDAR 3D depth points. This highlights the calibration inconsistencies in KITTI dataset, particularly after the initial day of data.

**Scale Alignment:** In Table 1, we compared three monocular depth estimation models by calculating the ratio between model-predicted depths and both ground truth and physics depth. Results show the scaling factor derived from physics depth closely matches that of the ground truth, with strong performance in the Monovit model. This indicates that physics depth can reliably replace LiDAR for calculating the scaling factor, enhancing the autonomy of self-supervised models.

## 5.2 EVALUATION OF PHYSICS DEPTH

In this paper, we systematically generated physics-based depths for the entire KITTI and Cityscapes datasets to support model training. We examined variations in road and flat surface physics depths across both datasets. As indicated in Tables 5, around 90% of KITTI pixels had errors below 10%, with 80% showing less than 5% error compared to LiDAR depths. The Cityscapes dataset performed even better, with 95% of pixels within 10% error and 85% within 5% compared to the Cityscapes disparity data. While road physics depth exhibited higher accuracy than flat surface depth, road pixels were fewer in number. To increase pixel density, we extended the physics depth approach to flat surfaces, though this introduced slightly larger error margins. Nonetheless, as shown in Tables 5, despite being less accurate, the flat surface depth still enhances the dataset and helps mitigate overfitting. Our analysis also revealed that KITTI had lower accuracy than Cityscapes, likely due to differences in camera calibration—KITTI uses one calibration file per day, whereas Cityscapes provides individual calibration files for each image. This suggests that improved calibration con-

tributes to better physics depth accuracy. Our method, particularly for flat surfaces like roads, shows strong potential to replace LiDAR for calculating scale factors in self-supervised monocular depth estimation. Visual results are provided in Figure 2.

### 5.3 DEPTH ESTIMATION

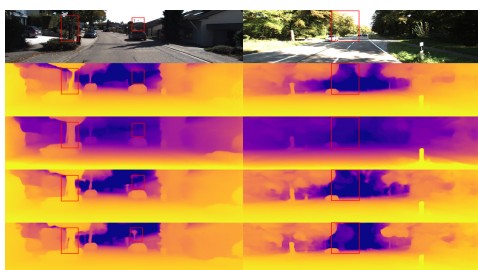

**KITTI:** We evaluated our model on the KITTI dataset. As shown in Table 3, our method outperforms previous methods. These gains are due to the integration of physics depth, confidence measures, and consistency checks in both 2D and 3D spaces. Figure 4 highlights the model's superior ability to capture detailed scene structures and achieve accurate reconstructions, surpassing ProDepth Zhang et al. (2024), Groco Cecille et al. (2024), , and Dual-Refine Bangunharcana et al. (2023).

Figure 4: Qualitative results on KITTI: From top to bottom the models are ProDepth Woo et al. (2024), Groco Cecille et al. (2024), DualRefine Bangunharcana et al. (2023), our models.

**Cityscapes:** We evaluated our model's generalization by fine-tuning and training it from scratch on the Cityscapes dataset, using a model pre-trained on KITTI for fine-tuning. As shown in Table 6, our model outperforms competing approaches.

**Make3D:** To assess generalization, we conducted a zero-shot evaluation on the Make3D dataset using the model pre-trained on KITTI. As shown in Table 7, our model achieves lower errors compared to other zero-shot models, demonstrating strong generalization capability. Figure 5 further illustrates the model's superior performance, delivering sharper depth predictions and more accurate scene details, showcasing its adaptability to diverse scenarios without requiring fine-tuning.

### 5.4 ABLATION STUDY

| Physic Depth | Contrastive Module | Camera Correction | AbsRel ↓ | Sq Rel↓ | RMSE↓ | RMSE log↓ | $\delta < 1.25$ ↑ |
|---|---|---|---|---|---|---|---|
| ✓ | | | 0.159 | 1.231 | 5.898 | 0.243 | 0.784 |
| ✓ | ✓ | | 0.090 | 0.641 | 4.170 | 0.183 | 0.895 |
| ✓ | ✓ | ✓ | 0.085 | 0.583 | 3.670 | 0.158 | 0.922 |

Table 4: Ablation study on KITTI: Ground depth represents the depth obtained using only the ground, while physics depth represents the depth obtained using the complete physics depth.

**Physics Depth:** Table 4 shows that using the complete physics depth results in smaller errors compared to using only ground depth. This suggests relying solely on ground depth may lead to overfitting, even though some areas of the physics depth may have errors.

**Contrastive Loss:** Table 4 indicates that the contrastive learning loss function can fully utilize both accurate and inaccurate depth values within the physics depth map and mitigate the impact of inaccurate physics depth on the model. This improves the accuracy of depth estimation.

**Camera Calibration Correction :** Table 4 demonstrates that the corrected parameters significantly enhance the model's accuracy compared to the uncorrected calibration parameters. This suggests that the corrected rotation matrix enables more precise computation of physics depth, which, in turn, improves the model's overall performance.

## 6 CONCLUSION

In this paper, we propose a self-supervised monocular depth estimation model based on calculating physics depth using the camera model. Although existing self-supervised techniques show potential, they still lag behind supervised methods in terms of accuracy and often require ground truth to resolve scale issues. For physics depth, we also designed a Anchor-Target network model that can fully utilize both the correct and erroneous depth information, effectively enhancing the performance of self-supervised models. By leveraging physics depth, we resolve the scale problem in monocular depth estimation. Leveraging the physics depth priors in Anchor-Target contrastive learning setting to support depth estimation training and inference, depth estimation accuracy can also be significantly improved through an unsupervised learning scheme.

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

# A APPENDIX

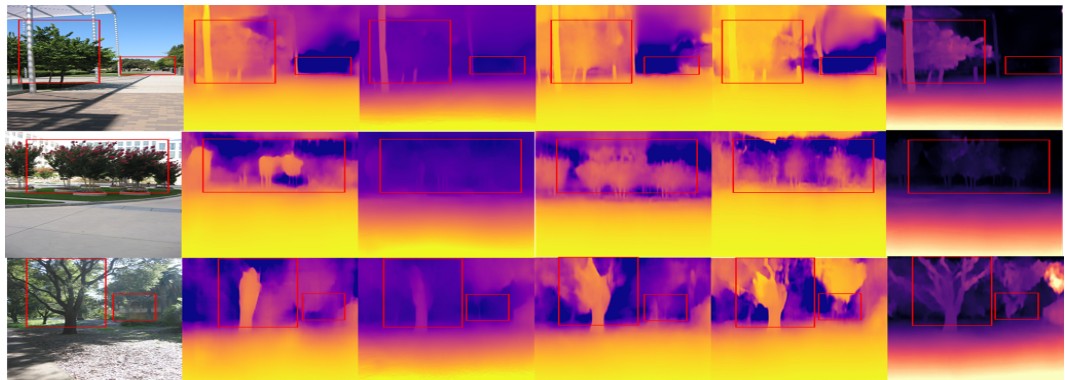

Figure 5: Qualitative results on make3d (Zero-shot): From left to right the models are ProDepth Woo et al. (2024), Groco Cecille et al. (2024), DualRefine Bangunharcana et al. (2023), SQLDepth Wang et al. (2023b), our models.

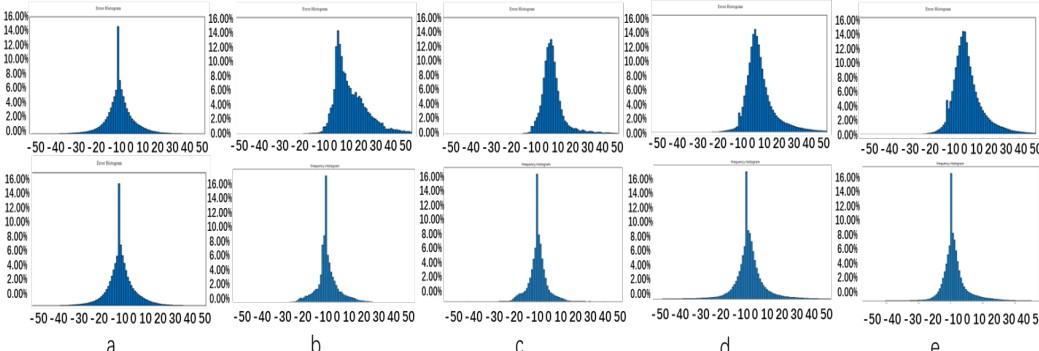

Figure 6: Percentage error frequency distribution for the KITTI. The first row highlights the distribution of percentage errors for road pixels, allowing for a comparison between physics depth and KITTI's LiDAR measurements at specific dates: September 26, 28, 29, 30 and October 3 of 2011. The second row shows the error distributions post adjustments to KITTI's camera calibration.

## A.1 LIMITATIONS AND FUTURE WORK

**Limitations:** Our current physics-based depth estimation method relies on modeling the scene as a flat ground plane and computing depth using known camera parameters. This assumption works well in structured outdoor environments, such as roads or open ground areas. However, in many real-world scenarios where the ground surface is not ideally planar, accurate physical priors become difficult to obtain, thus limiting the generalizability of the proposed method. **Future Direction:** To address this limitation, future work will explore slope-aware modeling by estimating the inclination of the ground surface. By integrating local slope estimation with the camera-to-ground geometric relationship, we aim to extend physics-based depth estimation from flat planes to non-planar terrains, enabling more accurate and robust supervision in complex environments.

## A.2 EXPERIMENT

| KITTI Date Cityscape City | Road Physics Depth Error: +/- 5% | Road Physics Depth Error: +/- 10% | Ground Surface Physics Depth Error: +/- 5% | Ground Surface Physics Depth Error: +/- 10% |
|---|---|---|---|---|
| 2011-09-26 | 84.28% | 96.26% | 75.08% | 89% |
| 2011-09-28 | 80.61% | 85.64% | 61.21% | 77% |
| 2011-09-29 | 90.53% | 97.34% | 74.46% | 91% |
| 2011-09-30 | 76.43% | 91.86% | 56.98% | 81% |
| 2011-10-0 | 78.12% | 94.61% | 62.77% | 85% |
| aachen | 87.48% | 94.77% | 73.17% | 86.94% |
| bochum | 80.76% | 93.22% | 65.51% | 83.95% |
| bremen | 86.55% | 97.64% | 72.60% | 88.29% |
| cologne | 81.66% | 98.88% | 75.14% | 88.82% |
| darmstadt | 82.49% | 95.44% | 69.95% | 86.56% |
| dusseldorf | 83.22% | 93.59% | 68.79% | 84.96% |
| erfurt | 83.78% | 94.26% | 69.58% | 85.85% |
| hamburg | 82.77% | 96.81% | 67.93% | 84.22% |
| hanover | 76.59% | 97.45% | 64.71% | 83.00% |
| monchengladbach | 83.42% | 94.73% | 63.75% | 82.48% |
| strasbourg | 84.63% | 95.62% | 61.44% | 81.52% |
| stuttgart | 80.49% | 96.38% | 68.52% | 85.26% |
| tubingen | 85.44% | 92.76% | 67.22% | 84.69% |
| ulm | 89.00% | 98.38% | 73.35% | 87.89% |
| weimar | 80.06% | 93.69% | 64.47% | 82.58% |
| zurich | 88.99% | 97.52% | 70.72% | 85.82% |
| jena | 77.90% | 92.85% | 63.75% | 81.85% |
| krefeld | 86.23% | 94.11% | 65.83% | 83.92% |

Table 5: **Comparison of error between physics depth and ground truth.** This table compares the error between physics depth and ground truth in the KITTI and Cityscape. The proportion of road and ground physics depth error within 5% and 10% of the ground truth for different days and cities.

| Method | Size | Test | AbsRel ↓ | Sq Rel↓ | RMSE↓ | RMSElog ↓ | $\delta < 1.25$ ↑ | $\delta < 1.25^2$ ↑ | $\delta < 1.25^3$ ↑ |
|---|---|---|---|---|---|---|---|---|---|
| Pilzer et al Pilzer et al. (2018) | $512 \times 256$ | 1 | 0.240 | 4.264 | 8.049 | 0.334 | 0.710 | 0.871 | 0.937 |
| Struct2Depth Casser et al. (2019) | $416 \times 128$ | 1 | 0.145 | 1.737 | 7.280 | 0.205 | 0.813 | 0.942 | 0.976 |
| Monodepth2 Godard et al. (2019) | $416 \times 128$ | 1 | 0.129 | 1.569 | 6.876 | 0.187 | 0.849 | 0.957 | 0.983 |
| Lee Lee et al. (2021b) | $832 \times 256$ | 1 | 0.111 | 1.158 | 6.437 | 0.182 | 0.868 | 0.961 | 0.983 |
| InstaDM Lee et al. (2021a) | $832 \times 256$ | 1 | 0.111 | 1.158 | 6.437 | 0.182 | 0.868 | 0.961 | 0.983 |
| ManyDepth Watson et al. (2021) | $416 \times 128$ | 2 | 0.114 | 1.193 | 6.223 | 0.170 | 0.875 | 0.967 | 0.989 |
| SQLDepth Wang et al. (2023b) | $416 \times 128$ | 1 | 0.110 | 1.130 | 6.264 | 0.165 | 0.881 | 0.971 | 0.991 |
| Ours | $416 \times 128$ | 1 | 0.103 | 1.090 | 5.937 | 0.157 | 0.895 | 0.974 | 0.991 |

Table 6: The quantitative depth comparison of the Cityscape dataset. M: trained with monocular videos; MS: trained with stereo pairs.

| Method | Type | AbsRel ↓ | Sq Rel↓ | RMSE↓ | log10↓ |
|---|---|---|---|---|---|
| Zhou Zhou et al. (2017) | S | 0.383 | 5.321 | 10.470 | 0.478 |
| DDVO Wang et al. (2018) | M | 0.387 | 4.720 | 8.090 | 0.204 |
| Monodepth2 Godard et al. (2019) | M | 0.322 | 3.589 | 7.417 | 0.163 |
| CADepthNet Yan et al. (2021) | M | 0.312 | 3.086 | 7.066 | 0.159 |
| SQLDepth Wang et al. (2023b) | M | 0.306 | 2.402 | 6.856 | 0.151 |
| Ours (Backbone: SQLDepth) | M | 0.304 | 2.213 | 6.792 | 0.148 |

Table 7: The quantitative depth comparison of the Make3D.

