# OpenReview forum: "Enhancing Self-Supervised Depth Estimation Through Camera Parameter Priors"
_ICLR.cc/2026/Conference — ICLR 2026 Conference Withdrawn Submission_

### Official Review · Reviewer_AQg2 · 2025-10-30

**Soundness:** 2
**Presentation:** 2
**Contribution:** 2
**Rating:** 2
**Confidence:** 3

**Summary:**

The paper propose a method to train a model that predicts depth map from a single image at metric scale. Real-world camera heights are assumed to be known during training and is used to recover metric depths, which are then used as pseudo label ground-truth depth to supervise another student network. As a results, the method predict depth at metric scale and achieve comparable performance as if Lidar depth are used to recover the true depth scale.

**Strengths:**

- The paper is well-written and easy to follow.
- The paper can predict depth at real-world scales.

**Weaknesses:**

- Novelty of the method: The core contribution of the method is (1) recovering metric scales using real-world measurement of the camera height, and (2) distill knowledge to a student monocular depth estimation network. These two modules are not new and have been presented in previous works. Particularly, (1) the scale recovery method using camera height has been introduced in [1+]  , while (2) has been introduced in [2+] which measure absolute depth from multi-view images, then used as pseudo ground-truth to supervise a student network.

- It is unclear what is the goal of the camera calibration correction methodology. Does the network also predict the camera poses? I would appreciate if the authors could provide clarification on this.

- Line 196 mentions that "For objects not touching the ground, depth is extrapolated from nearby objects." However, it is also unclear how to define whether the object touch the ground or not, and how to determine nearby objects? What if the target object does not have any object close to it?

[1+] Xue, F., Zhuo, G., Huang, Z., Fu, W., Wu, Z. and Ang, M.H., 2020, October. Toward hierarchical self-supervised monocular absolute depth estimation for autonomous driving applications. In 2020 IEEE/RSJ International Conference on Intelligent Robots and Systems (IROS) (pp. 2330-2337). IEEE.
[2+] Shyam, P., Okon, A. and Yoo, H., 2024. Enhancing Self-Supervised Monocular Depth Estimation via Piece-Wise Pose Estimation and Geometric Constraints. In Proceedings of the IEEE/CVF Winter Conference on Applications of Computer Vision (pp. 231-241).

**Questions:**

- The method figure looks distorted, making it difficult to read the texture within the figures.

---

### Official Review · Reviewer_CCzv · 2025-10-30

**Soundness:** 3
**Presentation:** 3
**Contribution:** 2
**Rating:** 4
**Confidence:** 4

**Summary:**

This paper addresses scale ambiguity and under-utilized camera physical information in self-supervised monocular depth estimation. It proposes a framework that uses physics-derived depth priors—computed from camera intrinsics/extrinsics and semantic cues—to provide metric-scale supervision. A partial "physics depth" map (e.g., ground, aligned structures) is calculated via camera projection and filled to form a dense prior, offering absolute scale (unlike conventional relative scale).This prior integrates into a two-network scheme: an Anchor (teacher) network trained with physics depth as pseudo-labels, and a Target (student) network updated via exponential moving average, learning from photometric reconstruction and a novel contrastive loss. Pixels are classified by prediction entropy (low-entropy pixels are reliable, high-entropy pixels are unreliable): reliable pixels supervise the Target, while unreliable ones act as contrastive loss negative samples.The method outperforms state-of-the-art models on KITTI (no external LiDAR), generalizes to Cityscapes (finetuning/scratch training) and zero-shot Make3D. Key contributions include camera physics-based supervision, scale ambiguity resolution, entropy-driven contrastive learning, and calibration correction for reliable physics depth.

**Strengths:**

Camera geometry supervision: Uses camera parameters and simple scene assumptions to compute physics-based depth priors, providing true metric scale without external sensors—creatively solving longstanding scale ambiguity.

Effective contrastive framework: Anchor-Target (teacher-student) network with entropy-based pixel screening uses all data: reliable estimates supervise the student, high-entropy predictions serve as hard negatives, maximizing information usage over discarding uncertain data.

Absolute scale & calibration: Outputs scale-correct depth (unlike standard self-supervised methods) and includes camera-to-ground calibration verification/correction, ensuring physics depth accuracy—critical for real-world deployment (e.g., robotics).

**Weaknesses:**

Technical detail clarity: Contrastive loss design is mentioned but underelaborated, e.g., how anchor/positive/negative samples are selected per class from unreliable pixels (presumed via depth-bin classification, but details are sparse).

Limited evaluation scope: All experiments use driving/outdoor datasets; performance under broken assumptions (e.g., significant camera roll, no flat ground) remains unclear.

**Questions:**

Semantic segmentation role: How is segmentation obtained/used? Is the Anchor network’s segmentation output supervised (ground-truth labels) or self-supervised? If supervised, is labeled data needed for new environments? How does segmentation error (e.g., misclassified road pixels) impact physics depth and training?

Contrastive loss details: How are negative samples selected/used? Are unreliable pixels negatives for all classes except their predicted one, or only the least likely depth bin? How does contrastive loss weight affect training stability (too high/low)?

Computational overhead: The framework is more complex than standard self-supervised models—can you comment on training time and memory overhead?

---

### Official Review · Reviewer_D1T7 · 2025-11-01

**Soundness:** 1
**Presentation:** 2
**Contribution:** 1
**Rating:** 2
**Confidence:** 4

**Summary:**

This paper is about utilizing camera physical model parameters to calculate scene depth, which the authors call "physics depth," to provide supervisory signals for depth estimation. They introduce a contrastive learning self-supervised framework designed to integrate this physics depth supervision, aiming to provide an absolute scale and enhance accuracy even when the generated priors are noisy.

**Strengths:**

I appreciate the authors' goal of addressing scale ambiguity in self-supervised monocular depth estimation. I like the core idea of trying to ground the depth estimation in the physical camera model (intrinsics and extrinsics). Deriving an absolute scale is a persistent issue in purely image-based methods, and using known geometry is a sensible direction.

**Weaknesses:**

I have significant concerns regarding the fundamental assumptions and the methodology, leading me to recommend rejection. The core claims of the paper are not well supported.

- Misleading Claim of Self-Supervision. The paper presents the method as "self-supervised," but this is fundamentally untrue. The entire pipeline depends on semantic segmentation to identify the ground plane to calculate the initial physics depth. The authors explicitly state that the semantic segmentation task being trained using supervised learning. Relying on expensive, manually annotated semantic labels means the framework is not self-supervised. This makes comparisons to actual self-supervised methods unfair and significantly undermines the paper's contribution.

- The entire "physics depth" calculation hinges on the assumption of a perfectly flat ground plane and known camera height/orientation relative to it (Section 3.1). This assumption is frequently violated in real-world driving scenarios. The method's applicability is severely limited by this idealized geometry. (The high accuracy reported might simply be overfitting to the relatively flat environments in the KITTI dataset.)

- The method for extending depth from the ground to the rest of the scene (Section 3.2) is highly heuristic and lacks geometric rigor. Propagating depth values to vertical objects assumes perfect verticality and connectivity. More concerning is the use of the Telea image inpainting technique (L195) to fill gaps. Inpainting algorithms are designed for visual texture consistency, not geometric accuracy. This introduces fundamentally unreliable supervisory signals. Table 2 validates this concern: accuracy drops drastically from the road surface to the final dense physics depth (approx. 52%). Training a network with labels where half the data is significantly erroneous is problematic.

- Reliance on LiDAR for Calibration Correction. The method requires accurate extrinsic parameters. The authors note inconsistencies in the KITTI extrinsics and propose a correction (Section 3.3). However, this correction process relies on existing LiDAR data to compute the ground plane normal and the rotation matrices. If LiDAR is needed to calibrate the setup to generate the supervisory signal, it contradicts the goal of moving away from reliance on such sensors.

**Questions:**

- How sensitive is the physics depth calculation to dynamic changes in camera extrinsic parameters? In real driving conditions, height and pitch fluctuate due to vehicle suspension dynamics (e.g., hard braking). Was this analyzed?

- The definition of the sky depth as "1.5 times the maximum inpainted depth" seems completely arbitrary. What is the geometric justification for this choice?

---

### Official Review · Reviewer_tGeP · 2025-11-01

**Soundness:** 3
**Presentation:** 3
**Contribution:** 3
**Rating:** 6
**Confidence:** 4

**Summary:**

This paper proposes to enhance self-supervised monocular depth estimation by introducing a physics-based depth prior computed from known camera parameters (intrinsics and extrinsics). The authors claim this prior provides absolute scale and can be integrated into a contrastive self-supervised training framework without relying on LiDAR. Experiments on KITTI, Cityscapes, and Make3D show improved metrics compared to existing self-supervised baselines.

**Strengths:**

Strengths:
1. The integration of camera geometry priors into self-supervised learning is conceptually appealing and provides a meaningful link between physical modeling and data-driven training.
2. The contrastive Anchor–Target scheme shows careful engineering and a reasonable mechanism for managing unreliable pseudo-labels.
3. Experimental results convincingly demonstrate improvements within the automotive domain, with well-designed ablations and robust comparisons on common benchmarks.
4. The discussion of calibration correction and entropy filtering reflects solid implementation insight and awareness of real system deployment challenges.

**Weaknesses:**

Weakness
The entire method critically depends on the planar ground assumption, which severely limits its validity and undermines its claimed generality. The so-called physics depth prior is computed by projecting scene points onto a plane estimated as the road surface. This assumption is valid only for highway-like datasets such as KITTI but fails in any 3D structure that violates local planarity (e.g., hills, ramps, curves, indoor or off-road scenes). Without this assumption, the “physics prior” collapses — the model cannot compute valid pseudo-depths, meaning the approach loses its supervisory signal altogether. As a result, the method’s “physical prior” is not a generalizable learning principle, but a handcrafted geometric constraint tuned to a specific dataset bias. Thus, while the empirical results appear strong, they arise from the model being perfectly matched to its assumption-dominated dataset, not from a general or scalable self-supervised framework. The claim of achieving “absolute-scale depth estimation” without LiDAR does not hold outside of planar driving scenes.

Other questions.
1 How would the method behave under small variations in camera pitch or calibration noise (e.g., ±2° deviation)?
2. Since “calibration correction” relies on pre-existing LiDAR depth to compute daily offset, 3. could this introduce train–test information leakage?
3. Have the authors tested whether the pseudo-depth computation remains numerically stable when h or tan⁡(θ) are close to zero?

The paper is well executed and practically relevant for constrained driving scenarios, but its main idea is too tied to a restrictive geometric assumption.

**Questions:**

na

---

### Note · Authors · 2025-11-16

I have read and agree with the venue's withdrawal policy on behalf of myself and my co-authors.